# A Review on Properties and Application of Bio-Based Poly(Butylene Succinate)

**DOI:** 10.3390/polym13091436

**Published:** 2021-04-29

**Authors:** S. Ayu Rafiqah, Abdan Khalina, Ahmad Saffian Harmaen, Intan Amin Tawakkal, Khairul Zaman, M. Asim, M.N. Nurrazi, Ching Hao Lee

**Affiliations:** 1Laboratory of Biocomposite Technology, Institute of Tropical Forestry and Tropical Products, Universiti Putra Malaysia, Selangor 43400, Malaysia; ayu.rafiqah@yahoo.com (S.A.R.); harmaen76@gmail.com (A.S.H.); 2Engineering Faculty, Universiti Putra Malaysia, Serdang 43400, Malaysia; intanamin@upm.edu.my; 3Polycomposite Sdn Bhd, 75-2 Jalan TKS 1, Taman Kajang Sentral, Kajang 43000, Malaysia; dr.khairulz@gmail.com; 4Center for Defence Foundation Studies, National Defence University of Malaysia, Kem Sungai Besi, Kuala Lumpur 57000, Malaysia; zaidharithah66@yahoo.com

**Keywords:** poly butylene succinate, copolymers, biodegradability, mechanical, physical properties

## Abstract

Researchers and companies have increasingly been drawn to biodegradable polymers and composites because of their environmental resilience, eco-friendliness, and suitability for a range of applications. For various uses, biodegradable fabrics use biodegradable polymers or natural fibers as reinforcement. Many approaches have been taken to achieve better compatibility for tailored and improved material properties. In this article, PBS (polybutylene succinate) was chosen as the main topic due to its excellent properties and intensive interest among industrial and researchers. PBS is an environmentally safe biopolymer that has some special properties, such as good clarity and processability, a shiny look, and flexibility, but it also has some drawbacks, such as brittleness. PBS-based natural fiber composites are completely biodegradable and have strong physical properties. Several research studies on PBS-based composites have been published, including physical, mechanical, and thermal assessments of the properties and its ability to replace petroleum-based materials, but no systematic analysis of up-to-date research evidence is currently available in the literature. The aim of this analysis is to highlight recent developments in PBS research and production, as well as its natural fiber composites. The current research efforts focus on the synthesis, copolymers and biodegradability for its properties, trends, challenges and prospects in the field of PBS and its composites also reviewed in this paper.

## 1. Introduction

The expansion of industry produced not only many products for human activities but also a large amount of plastic waste to the environment, because used products are discharged after human activities. Plastics are polymers that are chemically synthesized from petroleum products that contain long chains of monomer. The plastics industry has been around for 60–70 years. Plastics have excellent properties in terms of flexibility, longevity, light weight, and low cost [1]. Plastic was invented in the 1860s, but it was not until the 1920s that it was produced for commercial use. In the 1940s, its production exploded, making it one of the world’s fastest-growing industries. Plastics expanded at an annual rate of 8.7% on average from 1950 to 2012, from 1.7 million tons to nearly 300 million tons today. In 2015, the world plastic increased to 8.3 billion tons and produced up to 76% plastic waste [2,3]. In 2018, it was showed that the annual growth rate of 8.4% of plastic production was about 360 million tons. It is estimated to reach 500 million tons in 2025. From this total production, 60% entered the environment as plastic waste [4].

Recent data revealed that the increased use of plastic materials in domestic and industrial sectors have exceeded their global production by up to 400 Mt/year, which poses serious concerns, mainly for their disposal, environmental contamination, toxicity to the ecosystem and human health [5]. Synthetic plastics produced a large fraction of submerged waste in the natural habitat and are considered as emerging pollutants with a significant impact on the environment owing to their large concentration, widespread distribution, and non-biodegradable property. There is significant evidence of plastic pollution in the aquatic ecosystem, including plastic islands and micro plastics [6]. Polymer has begun to replace advanced materials in recent years due to its superior properties. However, disposable materials account for more than a third of plastics production, resulting in waste and leading to plastic emissions [7]. The use of non-biodegradable polyethylene (PE) mulch films in greenery fields or soil has created serious problems in Southeast Asia [8]. Non-biodegradable polymers, i.e., polyethylene (PE), polypropylene (PP), ethylene vinyl alcohol, poly(ethylene terephthalate), polystyrene, expanded polystyrene, polyamides, polyurethane, and poly(vinyl chloride), have dominated in the packaging application because of their good physical and mechanical properties The disposal of synthetic plastics makes an additional contribution to the carbon dioxide emissions in the atmosphere, which contributes to global warming. The most obvious form of pollution associated with plastic packaging is waste plastic sent to landfills. Plastics are very durable and therefore remain in the atmosphere for a long time after they have been discarded, particularly if they are sheltered from direct sunlight by being buried in landfills [9]. In response to these problems associated with plastic waste, this article used PBS as the main biodegradable polymer to be discussed, due to the fact that there is rarely research on these materials for commercial application. The melting point for PBS is 115 °C, which is lower compared with polylactic acid (PLA). These properties could save on industrial processing as it takes a shorter time to melt and blends with other material. PBS is also easy to process and handle while mixing it with other materials. In terms of brittleness, PBS is more rigid but slightly brittle due to the fact that it is produced from petroleum-based materials. PBS also has good thermal stability and excellent mechanical properties [10]. Other bio plastic such as polyhydroxyalkanoates (PHAs) and polyhydroxybutyrate (PHB) also possess great properties. This bio plastic production contributes to between 25% to 30% of the total plastics market worldwide [11]. The biodegradability of PBS is an attractive attribute for single-use food packaging since it is able to degrade at high rates over short periods of time and has also been listed as being certified as compostable according to the Biodegradable Products Institute, and is available in direct food contact grades [12]. In 2003, Mitsubishi Chemicals built a 3000 tons/year capacity plant and launched to the market a PBS named GS Pla (Green and Sustainable Plastic). This polymer has high molar masses without the use of a chain extender. Since then, several PBS producers such as Hexing Chemical (Anhui, China), Xinfu Pharmaceutical (Hangzhou, China) or IRe Chemical (South Korea) appeared on the market. In 2010, Hexing Chemical became China’s first large-scale PBS enterprise, with the annual capacity of 10,000 tons. The same year, Xinfu Pharmaceutical announced the construction of the world’s largest continuous PBS production line, with an annual capacity of 20,000 tons [13]. Figure 1 shows increasing trends of bio-based polymer production in the year 2011 to 2020.

Mulching films occupy more than 8000 km^2^ of agricultural land, resulting in many negative effects but still preserving horticulture [15]. As a result of increased understanding of environmental contamination, people have been searching for polymers that can replace petroleum-based plastics [16]. As a result, biodegradable plastics have been produced to substitute non-biodegradable plastic in order to resolve the environmental problem [17]. The production of bio-plastic has made a major contribution to the mitigation of global warming [18]. The initiative to develop new biodegradable polymers has been made using renewable resources as the raw material. This is due to the rapid depletion of fossil fuels and the vast amount of non-biodegradable plastics in use. Green polymers, in particular, are an important alternative as a new eco-friendly material for growing the plastic waste material that is accumulated every day on Earth [16]. The growing market for biodegradable polymers such as polylactic acid, cellulose-based thermoplastics, and other polysaccharide-based plastics aids in the reduction of pollution [17,19]. Biopolymer prices are rising every year. Business Communication Company (BCC) published a research report in 2014, estimating global bio plastic demand at more than 1400 kt (metric kilo tonnes). Nonetheless, it is estimated that demand will increase to around 6000 kt in 2019. Due to increased demand, global production is projected to hit more than 7.8 million tons in 2019 [19]. Many biodegradable polymers have recently been launched. These include polysaccharide polymers (cellulose, chitosan, chitin, and starch), protein-based polymers (gluten, ovalbumin, soy protein, casein, and others), and bacterial polymers such as poly(3-hydroxybutyrate) or (3-hydroxybutyrate-co-3-hydroxyvalerate) [20]. Biodegradable polymers have a lot of promise in biomedical and environmental applications. As a result, fundamental science and technology have centered on it. Poly(butylene succinate) or PBS has greater biodegradability, thermal properties, melt processability, and chemical resistance than other aliphatic polyesters. With these properties, it is a promising plastic material in the industry. Injection molded items, fibers, and films have all been made from it [21,22,23].

Since the price of PBS is expensive, many researchers have used other materials to substitute a partial percentage of PBS, such as using natural fibers. Oil palm fiber and tapioca starch are used as reinforcing materials to minimize the amount of PBS used and the cost of production in PBS-based products. Reinforcement products also assist in improving the material’s strength. Furthermore, the combination of biodegradable polymers reduces the total cost of the material while also changing the desired properties and degradation rates. In terms of the matter, as opposed to the copolymerization process with biodegradable polymer blending, the latter was found to be a much simpler and faster method of achieving the desired properties [24]. PBS brittleness can be strengthened via the copolymerization process. The presence of a compatibilizer or additive will increase the miscibility of a blend composite, which increases impact strength. The use of urethane composite as a compatibilizer is an example [25]. Meanwhile, natural fibers such as oil palm, flax, and jute can be combined to create a biodegradable composite that is both environmentally sustainable and recyclable. In addition, natural fibers involve the use of a compatibilizer or reinforcement to cross link the community, resulting in good properties for potential use [26].

However, some of its drawbacks, such as slow crystallization rate, low melt viscosity, and softness, have limited its processing capabilities and applications, especially in injection molding. PBS strength properties deteriorate due to a rapid crystallization reaction when combined with other materials [27,28]. The rheological properties of PBS have a major effect on its processability. Its viscosity will decrease as the sheer rate increases. The low melt viscosity makes PBS production more difficult. The lack of accessibility is a shortcoming of PBS, which used to hinder its extension. To boost its properties for different purposes, PBS must be modified and treated [29].

Thus, the objective of this contribution is to collect, analyze and compare the results collected by research on the use of PBS concerning its copolymer, physical, mechanical properties and its applications.

## 2. Poly(Butylene Succinate)

Poly(butylene succinate) has seen a rise in demand in recent years as a result of its promising sustainability and biodegradability. It was discovered that the production of bio-based plastics was 4.2 million tons in 2016, and that it is projected to rise by 45 percent by 2021 [30]. PBS (poly(butylene succinate)) was first used in 1993 as a biodegradable polymer and is still commonly used in industry. Mulching films, compostable bags, nonwoven sheets and garments, catering goods, and foams are only a few of the applications [31]. PBS was also listed as a bio plastic, raising general awareness of environmental problems caused by non-biodegradable and non-renewable plastics, as well as the rapid loss of fossil fuel supplies. There are biodegradable polymers manufactured from petroleum-based materials that are also known as green polymeric matrices [32]. 1,4 butanediol (BD) is a compound that is widely present in fossil fuels and available on the market. It is worth noting that succinic acid and BD can be derived not just from oil, but also by the fermentation method.

Few microorganisms have been studied to generate succinic acid using biotechnological processes in recent years, and these processes have shown good yields [33]. PBS degrades faster in soil than petrochemical plastic and may not be toxic to the environment. The involvement of microorganisms such as *Fusarium solani* may cause it to degrade. It has been documented that 39 bacterial strains from the Firmicutes and Proteobacteria classes can degrade polyhydroxybutyrate (PHB), polycaprolactone (PCL), and PBS.

Plastics’ biodegradability is closely connected to their substance, and the chemical and physical properties of plastics affect the biodegradation process. The surface conditions of polymers play an important role in the biodegradation process, for example, surface area, hydrophilic and hydrophobic properties [34]. However, the price of PBS is expensive compared with petrochemical-based plastic such as polystyrene (PS), polyamides (PAs) polyethylene terephthalate (PET) and polyethylene (PE). The price is higher due to the cost of processing that involves electrochemical processing and the condensation of succinic acid and 1,4 butanediol [24]. Table 1 shows several grades and properties of PBS that has been commercially available.

### 2.1. Structure

PBS is a versatile semi-crystalline polymer with a semi-crystalline structure. PBS is in high demand in sectors because of these characteristics. PBS has similar physical properties to polyethylene terephthalate [39]. PBS has strong elongation properties and can be used in a number of applications [40]. PBS has an ester group in its chemical structure, which degrades into low molecular weight polymers when exposed to water. As the temperature rises, the rate of PBS depletion rises as well [41]. The chemical structure of the repeat unit is –[O–(CH_2_)*_m_*–O–CO–(CH_2_)*_n_*–CO]_N_, as shown in Figure 2. The values of *m* and *n* were found to be 4 and 2, respectively [39]. PBS has α or β crystal polymorphs. The β structure can be found when the material is under strain [42]. However, softness and gas barrier properties are lacking, necessitating the use of PBS blended with other materials such as fillers to satisfy application specifications [43]. PBS provides a wide range of workability; the glass transition temperature should be below room temperature to allow for manufacturing in a variety of ways, including extrusion, injection molding, and thermoforming [33].

### 2.2. Synthesis

Figure 3 shows that PBS can be synthesized in various ways such as the polycondensation of succinic acid (or dimethyl succinate) and 1,4-butanediol whereby the monomers can be obtained from fossil-based or renewable resources. The benefit of this synthesis method is that it increases thermal and mechanical properties, as well as thermoplastic processability [30]. The most popular method of processing petrochemical succinic acid is the catalytic hydrogenation of maleic acid or its anhydrite.

Figure 4 illustrates how the hydrolysis of maleic anhydride to succinic acid starts. Maleic acid is made when one of the single bonds between carbon and oxygen is destroyed. Furthermore, by adding hydrogen, it splits the carbon–carbon double bond and completes the reaction to produce succinic acid [44].

Meanwhile, the fermentation process can also produce succinic acid. Since microorganisms are used in the biotechnological method to manufacture succinic acid, various microorganisms have been screened and tested in order to produce bio-based PBS. Figure 5 illustrates how succinic acid can be converted to 1,4-butanediol via a hydrogenation process to create PBS [33].

By using renewable resources to produce succinic acid, it can be costly compared to the petroleum-based processes. Recent studies used different types of microorganisms for succinic acid production such as *Anaerobiospirillum succiniciproducens*, *Actinobacillus succinogenes* and *Mannheimia succiniciproducens* [47]. Nevertheless, due to high glucose concentrations and exposure to air, the organism experienced instability and deterioration [48]. Besides that, chemically synthesized aliphatic polyesters with high molecular weights were also able to improve properties of PBS. From previous studies, it was reported that the synthesis of poly(butylene succinate-*co*-ethylene succinate) through direct polycondensation reacts with diols and diacids in the existence of a powerful substances [49]. Referring to Figure 6, the reaction of PBS through the direct polycondensation process using N35 as a catalyst is shown [50]. PBS from Showa Highpolymer Co., Ltd. (Tokyo, Japan) was made with organometallic catalysts at high reaction temperatures (190 °C), resulting in a high molecular weight and the complicated removal of remaining metals or materials due to solid metal–ester interactions. In fact, the reaction of high temperature with organometallic compounds that occurs may be one of the causes of monomer/polymer decomposition reactions. As a consequence, the substance would discolor and have a lower molecular weight [51]. In addition, microwave irradiation can be used to synthesize PBS. This process is a safe, affordable, and an easy means of heating that also results in higher yields and shorter reaction times. According to previous studies, 1,3-dichloro-1,1,3,3-tetrabutyldistannoxane was used as a catalyst in the synthesis of PBS using this process [52].

### 2.3. Copolymers

Many tests of copolymerization with energy and good properties have been carried out. When two different forms of monomers are mixed in the same polymer chain, the result is a copolymer [53]. To achieve controlled enzymatic degradation or mechanical properties, poly(butylene succinate) can be mixed or blended with other monomers. As PBS is mixed with copolymers, the rate of degradation increases and the degree of crystallinity decreases. It was discovered that in order for cocrystallization to occur, the two crystallizable units in each crystal lattice must be compatible. Isomorphism happens where the elements of a copolymer have the same chemical composition, which increases the material’s properties [54]. In 1990, the Showa Denka company in Japan established various types of PBS copolymer such as poly(butylene succinate) (PBSu), poly(butylene succinate-co-butylene adipate) (PBSA) copolymer and poly(ethylene succinate) (PESu), shown in Figure 7, that were produced through a polycondensation reaction of glycols with aliphatic dicarboxylic acids and their secondary [55]. In the bulk, the copolymer is formed by a transesterification reaction. At 200 °C, an effective catalyst, tetra-n-butyl-titanate Ti(OBu)4, is used to manufacture high molecular weight aliphatic copolyesters from dimethyl esters of adipic, succinic acid, and 1,4-butanediol [49].

Figure 8 illustrates how polybutylene azelate was made from a two-stage melt polycondensation of the right mixture of dicarboxylic acids plus 1,4-butanediol. Titanium tetrabutoxide was also used as a catalyst. It was first conducted in a nitrogen atmosphere at 150 °C for 6 h, then in vacuum at 180 °C for 18 h (5 Pa). Precipitation with ethanol of chloroform solutions disinfected the polymers (10 wt percent) [56].

Acidolysis, alcoholysis, and transesterification are three of the intermolecular reactions involved in the copolymer reaction, as shown in Figure 9. The thermal properties of PBS can be enhanced as it is blended with polycarbonate/poly(butylene terephthalate) (PBT), since PBT has a lower glass transition temperature and melting temperature, and blending with other materials creates structural differences in the polymer, such as sequence distribution [47]. Lastly, synthesis of the aliphatic copolyesters of poly(1,4-cyclohexanedimethanol-co-isosorbide 2,5-furandicarboxylate) was synthesized by two stages, which are esterification and polycondensation, as shown in Figure 10. This reaction was able to enhance thermal properties such as melting temperature (T_m_) and glass transition temperature (T_g_) compared with neat PBS [57]. Besides, Table 2 listed some of the major used of copolymers blend with PBS and their blended properties.

### 2.4. Treatments of Poly(Butylene Succinate)

PBS is a biodegradable polymer of excellent properties. However, it is porous and has poor density and flexibility. As a result, researchers are exploring surface alteration on PBS in order to build polymeric materials with biocompatible and bioactive surfaces. The surface alteration was achieved by injecting a gas such as H_2_O or NH_3_ into the chamber to produce plasma. This therapy increases the hydrophilicity of PBS and is also known as plasma treatment [61]. Surface polymerization, surface immobilization of biocompatible molecules, and plasma-based approaches are just a few of the techniques that can be used to alter the surface of polymeric biomaterials. Plasma immersion ion implantation (PIII) was selected as the best approach because of its ease of use and long-term results [62]. Plasma treatments are efficient finishing methods for polymers in industrial applications. Using a radio frequency generator, plasma was produced by inductively coupled discharges at 13.56 MHz, and the gas was injected into the PBS sheet surface [63]. Plasma cured polymers and their contacts with layers coated can be exposed to a number of surface chemical analyses. These studies have yielded knowledge ranging from basic chemical structure, which can be used to classify functional groups, to the precise recognition of bonding between atoms in a treated surface and those in a coating layer. Plasma therapies can be studied in a number of ways, and their impacts on polymer surfaces can be analyzed using a variety of techniques. Some are interested in the characterization of functional plasma sources, while others are interested in the characterization of plasma-treated structures [64]. According to other studies, PBS layer surfaces became more wettable after being treated with continuous plasma or pulsed plasma, but they did not find any signs of biodegradation in the plasma-treated PBS sheets. Chemical graft polymerization is another treatment for PBS that is widely used to improve its surface hydrophilicity [65]. Aside from that, the photografting polymerization of hydrophilic acrylic acid (AA) and hydrophobic styrene (St) monomers changed the PBS surface. By increasing the reaction time of the photografting polymerization of AA, this modification assists in making the PBS surface more hydrophilic. This procedure was chosen because it was low-cost, had moderate reaction conditions, and resulted in lasting and stable chemical modifications to the surface [66].

## 3. Poly(Butylene Succinate)-Based Composites

### 3.1. Physical Properties

The existing physical properties of PBS was changed slightly due to the addition of other materials such as reinforcement, additive or filler. Much research that used PBS showed improvement in terms of the physical properties of the composite. The brittleness properties of PBS can be overcome by the addition of plasticizer such as glycerol which also can improve the elongation of the PBS for future application. The synthetic technique used to make PBS, which included copolycondensation, reactive blending, and physical blending, was connected to its physical properties. Aside from that, the degree of crystallinity influences the rate of PBS biodegradation [33]. The solid state structure or crystallinity of PBS can be identified using X-ray diffraction analysis [67]. According to previous studies, higher degrees of crystallinity cause lower hydrolytic and enzymatic degradation rates since amorphous domains that are less coordinated and packed are more easily targeted [68]. The degree of crystallinity of PBS between polyethylene (Tg −120 °C) and polypropylene (Tg −10 °C) which possess great properties and can proceed on polyolefin processing machines at temperatures of 160–200 °C for various applications [69]. When PBS was blended with other components with varying melt viscosities, the morphology of blended materials reveals a homogeneous dispersed phase of the product with lower viscosity [43]. The homogenous dispersed phase in the blends led to good properties for the composite in the future application. However, an addition of additive and binding agents was also required to achieve the desired properties of the composite. Other than that, research on PBS blended with lignin showed an increase in the melt flow index and the density of the composite when there was an increase in the percentage of lignin in the composite blends [70]. This means that in future applications, the composite could allow for the addition of reinforcements such as natural fibers without the loss of processability.

Another analysis observed cavities in PBS/agro flour composites, which may have led to poor stress transfer from the matrix polymer to the agro flour, resulting in poor tensile properties [71]. Using the Binder K720 Climatic Chamber (USA) at 30 °C, 60 percent, and 90 percent relative humidity (RH), a study on water absorption was performed. The water absorption improved as the proportion of organo-montmorillonite (OMMT) increased due to the hydrophilic nature of the octadecylamine (ODA) groups in the composite of PBS with organo-montmorillonite (OMMT). Figure 11 indicates that moisture absorption curves at 60 percent, 90 percent, and 100 percent relative humidity all followed the same pattern. The association of water molecules with the ODA groups of OMMT resulted in the creation of a hydrogen bond [72]. Owing to the amount of hydroxyl groups and high hydrophilicity, the physical properties of jute fiber reinforced PBS composite increased water absorption as the fiber content increased [73]. From other research, it was found that the mass molar weight of PBS/curaua fiber composite is highest compared with sisal, coconut and bagasse. The water absorption percentage of curaua fiber composite reached 1.7 wt%, 4.6 wt% for P-bagasse, 3.4 wt% for P-coconut, and 2.6 wt% for P-sisal fibers. The high water absorption by bagasse and coconut was related to the lower crystallinity of fiber meanwhile, for curaua and sisal there was more resistance to water due to higher lignin content [74]. Natural fibers are hydrophilic in nature and consist of a moisture content of approximately 8–12.6% due to the presence of cellulose, hemicellulose and lignin [75]. Reported by Nabi Saheb [76], moisture content of the fibers can vary between 5% to 10%. Other research revealed molar mass of PBS/hemp fiber composite was increased significantly by an increase of 20% of hemp content due to the increased free volume of the mixture [77]. From morphology analysis of PBS/distillers grains (DGs), it was explained that the interfacial adhesion between DG and PBS is less sufficient, which may lead to deficient stress transfer between the PBS matrix and the reinforcing DG fillers [78]. The physical properties of jute fiber reinforced PBS composite improved water absorption as the fiber quality increased due to the sum of hydroxyl groups and high hydrophilicity [79]. Due to their hydrophobicity, glycidyl methacrylate-grafted poly(butylene succinate) and palm fiber (PF) composites (glycidyl methacrylate-grafted poly(butylene succinate) (PBS-g-GMA)/PF) composites showed a greater tolerance to water absorption than PBS/PF composites. The hydrophilic aspect of PF induced an increase in water absorption as the percentage of PF increased [80].

### 3.2. Mechanical Properties

The mechanical properties of PBS are the most critical factor when determining the industrial applicability of newly synthesized polymers. After reducing the block length, the mechanical properties of the PBS homopolymer were changed to minimize the elastic modulus, tension at break, and increase the elongation at break [81]. It has been documented that increasing the PBS content in blends with other polymers induces a decrease in Young’s modulus and tensile strength, resulting in strain hardening [82]. The blending of poly(glycerol sebacate) with poly(butylene succinate/dilinoleate) (PBS-DLA) multiblock copolymer led to a reduction in the elastic modulus and of the stress at break, however no yield was observed in all cases [83]. According to research on food packaging use, structural support is a key element that affects the occurrence of packaging defects or breakage. As compared to neat poly(butylene adipate-co-terephthalate), neat PBS with a molecular weight (Mw) of 1.4 × 10^5^ g mol^−1^ supplied by Showa Highpolymer Co. Ltd., Tokyo, Japan, demonstrated higher tensile yield strength but lower elongation (poly(butylene-adipate-co-terephthalate)—PBAT). During the tensile test, no strain hardening was found in PBS [60]. PBS composites blended with curaua fiber enhanced the impact strength as more curaua fiber was applied, and the flexural strength increased by nearly 64 percent as compared to the smooth polymer due to well-spaced fibers in the matrix [84]. Other researchers looked at PBS reinforced with jute fibers and found that it improved the tensile strength by 517.9%, tensile modulus by 3529.8%, flexural strength by 302.6%, and flexural modulus by 1949.1%. The composite’s mechanical properties were highest when 50 percent jute fiber material was applied [73].

Meanwhile, other studies on a mixture of PBS and cellulose acetate showed that the fracture stress was enhanced 1.5 times as compared to tidy PBS, and Young’s modulus rose from 320 MPa to 645 MPa [18]. As compared to pure PBS, the tensile strength, modulus, flexural strength, and modulus of PBS with coir fiber composite improved by 54.5 percent, 141.9 percent, 45.7 percent, and 97.4 percent. A good interface and sufficient fiber material in the composite have better wetting and improved mechanical properties [85]. Other experiments on mixing PBS with kenaf fiber indicated a 53 percent improvement in the tensile modulus, but the tensile strength and fracture strain of the composites decreased with increasing kenaf fiber content due to insufficient composite adhesion and void content [86]. The incorporation of organic or inorganic fillers modified the PBS properties drastically. Due to the composite’s improved stiffness, the elastic modulus and elongation at break increased and decreased with increasing filler material, respectively [87].

Other researchers discovered that the mechanical properties of cotton-reinforced PBS composites were studied using different cotton fiber loadings, as seen in Figure 12. The addition of 10–40% cotton fiber to PBS increased the tensile strength by 15–78%, but due to the brittleness of cotton fiber, the elongation rate decreased with increasing cotton fiber [8]. Other research on PBS reinforced with sisal fiber showed higher impact resistance compared to bagasse, coconut and curaua fiber [74]. However, as the fiber content of a PBS/palm fiber composite was increased, the tensile and flexural strength of the composite decreased due to low fiber dispersion in the PBS matrix and incompatibility between the two materials [80]. Inter-fibrillar voids are created as entangled fibers clump together, leading to a decrease in connections and surface area between the fiber and matrix [79]. PBS/microfibrillated cellulose composites showed that the tensile strength and modulus of composite fibers improved from 112 ± 10 MPa to 250 ± 16 MPa (123% increment) and 1.2 ± 0.04 GPa to 4.0 ± 0.10 GPa (233% increment) by increasing the fiber content [88]. According to other studies, applying distillers grains (DG) to composites improved their mechanical properties. The tensile modulus grew from 382.2 to 554.1 MPa, while the tensile power and elongation at break dropped from 35.1 to 15.2 MPa and 379 percent to 3.77 percent, respectively. The decline in elongation at break is most likely due to DG fillers in the matrix inhibiting PBS chain twining and negative interfacial adhesion [78]. Next, PBS/perennial grass composite showed an increase in the tensile strength at fiber loading 50 wt% at a modulus value of 3.88 GPa, which is 488% higher than that of neat PBS. However, adding fiber caused a reduction in the elongation rate due to the phase separation phenomena and reduced the polymer chain entanglement in the presence of rigid fibers [89]. Researchers, on the other hand, discovered a number of mechanical findings due to a variety of reasons including matrix and fiber incompatibility, incorrect processing methods, fiber deterioration, and others.

### 3.3. Thermal Properties

The glass transition temperature (Tg) and melting points of PBS decide its thermal behavior. The melting temperature of PBS blends containing hydroxyapatite and chitosan was lowered, and the degree of crystallinity increased significantly [90]. When the crystallization temperature increases, the amorphous thicknesses for PBS decreases and gives variation on its amorphous thickness [67]. Other researchers discovered that polymer crystallization at different temperatures results in different melting temperatures. The lamellar crystal structure and size can be the main cause of melting. The composition, chemical composition, and laboratory conditions all affect the lamellar crystal structure and size of copolymers [91]. A study on melting behavior of PBS could give one understanding of the polymer processing. A previous study showed the melting process of the cold-crystallized PBS and identified three peaks; a small endotherm (T_m_1) at 42 °C, a broad exotherm (T_ex_) at around 85 °C, and a large endotherm (T_m_2) with its maximal at 113 °C [92]. Pure PBS showed a sharp exothermal peak at temperatures of 92 °C and 114 °C. When more than 50% of the PBS composition was combined, the exothermal peaks appeared as single peaks with a shoulder. For PBS blends containing a high percentage of PBS, a higher crystallization rate and a lower crystallization temperature are likely. The thermodynamic stability of PBS (Bionelle 1020) supplied by Showa Highpolymer Co. Ltd. in Japan was measured at 300 °C. PBS degrades in a cyclic fashion, with anhydrides, olefins, carbon dioxide, and esters being the most important byproducts. From differential scanning calorimetry (DSC) analysis, the heat of fusion (∆H) of PBS/cotton fiber composite increases with an increase in fiber due to the nucleating effect of the fiber. The (∆H) of neat PBS was reported to be 110.3 J/g [8]. From previous research, it was reported that the (∆H) of a PBS/kenaf composite increased with the addition of kenaf fiber at a maximum 20% of fiber content. Neat PBS showed a low crystallization temperature (76.3 °C) in Figure 13, however, the crystallization peak of PBS shifted to a higher temperature with the addition of kenaf fiber in the composites due to the development of more crystals [86]. TGA examination of the PBS/cotton fiber composite revealed two peaks in Figure 14, the first at 345 °C, which corresponds to cellulose degradation, and the second at about 400 °C, which corresponds to PBS matrix decomposition [8]. The release of absorbed moisture in the kenaf fiber induces similar actions in PBS/kenaf composites that are in the first stage at 30 to 140 °C. The breakdown of cellulosic compounds such as hemicelluloses and cellulose happens in the second stage at 140–360 °C. The third step, which takes place between 360 and 500 degrees Celsius, is concerned with the deterioration of noncellulosic materials in the fiber [86]. Other studies using a PBS/bagasse composite showed thermal deterioration at 260–270 °C, weight loss at 50–60 °C due to moisture content loss, and a final decomposition stage at 400 °C [74]. Another study on PBS/bamboo husk composites found that integrating fiber into PBS increases its flame retardancy. According to DMA research, the homogeneity and compatibility of two materials together leads to improved mechanical properties [92]. The Tg of the PBS/hemp composite was decreased with an increase in the fiber content, because hemp does not contribute to the glass transition [78]. With rising nano- CaCO_3_ content, PBS/nanometer calcium carbonate (CaCO_3_) increased the thermal stability and improved the crystallinity index. CaCO_3_ particles assisted in the creation of carbide, which stopped the decomposed constituents from volatilizing [93].

## 4. Poly(Butylene Succinate)-Based Hybrid Composites

Hybrid composites are made up of two or more distinct types of fibers or fabrics that are mixed together in a typical or different form of polymer. The use of a combination of two forms of short fibers in a single polymer matrix has benefits over using either fiber alone [94]. Many researchers reported that hybridization may enhance the stiffness, strength, as well as the moisture resistance of the composite [95,96]. Many researchers in hybrid composites chose materials which were compatible and had desired properties such as sisal with oil palm fiber [97].

### 4.1. Physical Properties

Modified montmorillonite (OMMT) as a filler in blended polypropylene (PP)/PBS showed OMMT dispersion in blended polymers very well and also showed good interfacial reaction between PP and PBS blends. Besides that, the addition of OMMT also increased the viscosity blends at frequency (100–5 rad/s) [98]. The pore size distributions of TS-1 zeolite were calculated using the Barrett–Joyner–Halenda (BJH) model. The PBS/TS-1 zeolite hybrid composite showed uniformly globular morphologies with a median peak around 1.6–2.6 nm and a width of 80–100 nm. The TS-1 zeolite is evenly spread in the PBS matrix and has a radius of 100 nm [99]. XRD analysis of poly(propylene) (PP) and poly [(butylene succinate)-*co*-adipate] (PBSA) blended with organoclay indicated a higher degree of interaction, with the PBSA chains intercalating much of the silicate layers. The materials’ hybridization indicated strong compatibility and dispersion [100]. An analysis into short roselle and sisal fiber-reinforced hybrid polyester composites showed that increasing fiber content improved moisture absorption. It was also discovered that long fiber composites absorb more moisture than short fiber composites, affecting the percentage of moisture uptake into the composite [101]. The hydroxyl group of the kenaf/polyethylene terephthalate (PET) fiber reinforced polyoxymethylene (POM) hybrid composite reacted with the hydrogen bond of water molecules, resulting in high moisture absorption in the composite, according to a report. Virgin POM has a moisture content of just 0.2 percent, while hybrid POM has a moisture content of 6.7 percent. However, owing to the deterioration of cellulose and hemicellulose content, moisture absorption decreased from 6.7 percent to 0.55 percent when recycled kenaf composite was used [102]. Fiber has hydrophilic properties in nature that contribute by the hydroxyl group of cellulose and hemicelluloses constituents in the fiber cell walls [103]. Fibers contain 60–80 percent cellulose, 5–20 percent lignin, and up to 20% moisture, according to a previous study; however, different types of fibers can have different properties [74].

### 4.2. Mechanical Properties

Reported works on the study of mechanical properties of hybrid composite give an understanding on the strength, physical interaction between polymer and reinforcement and the combination of materials to form a hybrid composite [104]. Due to the homogeneous dispersion of PF in the PBS-g-GMA matrix, the mechanical properties of the glycidyl methacrylate-grafted poly(butylene succinate) (PBS-*g*-GMA) and palm fiber (PF) composite improved tensile and flexural strength [68]. Composites of poly(l-lactide) (PLLA)/poly(butylene succinate) (PBS) reinforced with organoclay (TFC) enhanced the tensile modulus from 1075 MPa to 1940 MPa; meanwhile, the elongation at break decreased from 72% to 5.3% as the content of TFC increased up to 10 wt% [105]. Other tests have shown that the tensile strength of PBS/TS-1 zeolite hybrid composites improves by 6–19% as compared to tidy PBS due to interfacial interaction between the TS-1 zeolite and PBS, which has a high surface energy. Then, at 2.0 wt percent of TS-1 zeolite particles, it began to decline due to the random agglomeration of zeolite particles [99]. It was also discovered that the tensile strength and modulus of pineapple leaf fiber (PALF)/recycle disposable chopstick hybrid fiber (CF)-reinforced PBS is higher than neat PBS. The tensile strength (32.8 MPa) is 2.2 times higher than that of neat PBS (14.8 MPa). Various fiber ratios were studied and it was found that 30 wt% fiber loading showed the best mechanical properties [106].

Due to its hydrophobic properties and high resistance to thermooxidative degradation, hybrid composites with higher resistance to environmental degradation can have lower tensile strength. Furthermore, the high cellulose and hemicellulose content of fiber induces high humidity, which can lead to low interfacial interaction between the polymer matrix and the environment [107].

Aside from that, fiber geometry may influence interfacial bonding between reinforcing fibers and the thermoplastic matrix, influencing the dynamics of load transfer between the fibers and the matrix [102]. When the percentage of fiber in the matrix is high, agglomeration occurs, which leads to efficient stress transfer between the fibers and matrix. However, increasing fiber content above a certain stage can cause dispersion problems, resulting in a reduction in mechanical properties [108]. Fiber–fiber adhesion decreases the contact area between fibers and the matrix in fiber bundles, resulting in poor tension transfer from the matrix process to the scattered fibers [79]. The tensile and impact strength of a PBS/lignin/switchgrass hybrid composite decreased as the filler material was increased. When compared to tidy PBS, the effect intensity was decreased by around 23% [109]. The incompatibility of two separate materials produces a decrease in tensile strength. It resulted in a low degree of interfacial adhesion between the hydrophobic polymer and the hydrophilic filler [69]. A lower storage modulus (tan) value suggests strong interfacial bond strength due to improved adhesion developed between the fiber and matrix [104].

### 4.3. Thermal Properties

The thermodynamic properties of different PBS hybrid composites were studied. With growing palm fiber (PF) material, DSC of glycidyl methacrylate-grafted poly(butylene succinate) PBS-*g*-GMA/PF composites revealed a decrease in melt temperature (*T*_m_) and heat of fusion (H) 81]. The decrease in (*T*_m_) was due to torque measurement and lower melt viscosity, causing the hybrid composite to be easily manufactured [75]. The addition of organoclay in a poly(l-lactide) (PLLA)/poly(butylene succinate) composite enhanced the thermal stability due to the fact that layers of organoclay are impermeable to small molecules generated during the thermal degradation process [105]. From TGA analysis, it was found that PALF/recycle disposable chopstick hybrid fiber (CF) in PBS effectively raised the char yields and heat of deflection from 17.5% to 33.6% [105]. Weight loss occurred between 340 °C and 370 °C due to the degradation of cellulose by depolymerization [110]. It was also stated that natural fibers undergo two stages of thermal degradation: first in the temperature range 220–280 °C for hemicellulose and another in the range 280–300 °C for the degradation of lignin [111].

## 5. PBS-Based Nanocomposites

Researchers in industry and academia have been involved in PBS-based nanocomposites because they often show substantial changes in material properties as opposed to pure polymer. Previous research has used carbon nanotubes (CNTs) combined with PBS as the focus of nanocomposite research, which has been shown to increase PBS thermal stability. Carbon nanotubes (CNTs) are a modern type of carbon made up of concentric cylinders of graphite layers. There are two types of CNTs: single-walled carbon nanotubes (SWNTs) and multi-walled carbon nanotubes (MWNTs). CNT was selected because of its excellent properties, which include a high aspect ratio, nanoscale diameter, low density, and, most specifically, excellent physical properties including exceptionally high mechanical power, high electrical and thermal conductivity [112]. Other experiments blended a PBS nanocomposite with organically engineered synthetic fluorine mica to create a PBS nanocomposite (OSFM). The mechanical properties of PBS were increased by about 120 percent in the value of the elastic modulus, and the stability of PBS was moderately increased in the presence of OSFM, according to this analysis [113]. Through integrating nanocomposites with biodegradable polymers such as PBS, researchers can boost the modulus, strength, heat resistance, gas permeability, and biodegradability. PBS/organically modified layered silicate (OMLS) demonstrates the increased degradability of nanocomposite relative to tidy PBS due to strong interfacial contact between the matrix and OMLS, according to previous studies [114]. According to other tests, OCMS nanocomposite has a substantial improvement in the tensile power, modulus, and biodegradability. Due to the strong barrier properties of the matrices after nanocomposites preparation, the biodegradation of PBS was improved [115]. Another analysis used a nanocomposite consisting of PBS and multi-walled carbon nanotubes (MWCNTs) that was made by melt-blending in a batch mixer. The mechanical properties of PBS were improved as a result of the analysis. The storage flexural modulus rose from 0.64 GPa for pure PBS to 1.2 GPa for the nanocomposite at room temperature. The elastic modulus has risen by approximately 88 percent. Furthermore, the electrical conductivity of neat PBS improved significantly after nanocomposite formation, from 5.8 × 10^−9^ S/cm for neat PBS to 4.4 × 10^−3^ S/cm for nanocomposite [116]. Solution intercalation, in situ intercalative polymerization, polymer melt intercalation, and template synthesis are some of the processes used to make PBS nanocomposite. Polymer melt intercalation, on the other hand, has been shown to be an outstanding procedure because of its simplicity, consistency with modern polymer manufacturing methods, and environmental friendliness due to the absence of any solvent, according to Yoshihiro et al. [117]. Aside from that, it was discovered that various cationic surfactants with functional groups and using polar functional groups such as maleic anhydride to the polymer matrix would enhance surface contact between polymer and silicate layer nanocomposite [118]. Researchers have documented advances in the PBS/Organomontmorillonite (OMMT) nanocomposite strength and modulus, as well as gas barrier and flame retardant properties. OMMT was also selected to create a “green” nanocomposite with the intention of using it in an environmentally friendly device [119].

### Biodegradability of PBS Composites

Biodegradable polymers such as PBS are easy to degrade in soil compared with synthetic polymers. The degradation of PBS is dependent on several factors including the molecular weight, water permeability, pH, temperature, purity, crystallinity, presence of hydroxyl or carboxyl groups, and catalytically acting additives that may involve bacteria, enzymes, or inorganic fillers [120]. PBS was first produced in the early 1990s by Showa Highpolymer, located in Japan. PBS is one of the biodegradable polymers that can be degraded by fungi and bacteria under natural conditions [121,122]. Researchers found that PBS can degrade by about 71.9% after 90 days and PBS in powder form is easier to degrade compared to the granule and film form. PBS uses 128 kJ/mol activation energy when decomposing, which is higher than poly(butylene terephthalate) (PBT) which only uses 117 kJ/mol energy [111,112]. Another study on PBS showed that copolymers are biodegradable in lipase solution, soil burial, water, activated sludge and compost. It started to degrade in water and CO_2_ naturally by enzymes [123].

Moreover, PBS also degrades through the hydrolysis process which occurs at ester linkages and reduces the polymer molecular weight [124]. From a previous study, a PBS/jute fiber composite showed the highest weight loss during a burial test; it is a composite with 10% fiber loading and the weight loss was around 62.5% compared to the neat PBS, which was only 31.4% [125]. Another study on PBS/rice husk flour (RHF) found that adding fibers into composite may enhance polymer surface thus increase the biodegradation rate of the composite. The temperature and humidity surroundings also influence the biodegradation rate [126]. A study on a PBS/abaca fiber (AF) composite also displayed that the biodegradation rate increased in the presence of fiber. PBS/AF was showed a 40% weight loss compared with neat PBS, which was only 7% after 90 days buried in soil. SEM analysis showed at 30 days the PBS/AF composite started to become rough and slowly degrade. Meanwhile, for neat PBS, the sample only starts to degrade at 90 days [127]. Other research on PBS/organo-montmorillonite (OMMT) stated the weight loss was consistent at 60 days and 90 days. The degradation rate was slow due to the improvement of the barrier properties which controlled the penetration of the microorganism through the material. The soil burial test has been widely conducted for evaluating the biodegradation activities of biodegradable polymers [128]. A similar observation was reported by [69] on the water permeability of composites also influenced the degradation rate because it depends on the transfer of water from the surface into the bulk. In a study, a PBS/Lyocell composite showed the highest weight loss of 75% at 60 days that was evaluated by the soil burial test. The direct degradation of the lyocell is caused by the microcrack or delamination of the composite [129,130,131]. Other researchers found that a PBS/Rubberwood (RBC) composite started to degrade at 60 days and showed crack and cavities on the composite. It is also mentioned that greater weight loss is due to low crystallinity that influences the biodegradation process. Some other factors that may influence the biodegradation rate is surface area, hydrophilic, hydrophobic properties, chemical structure and density [132].

## 6. Applications

Biodegradable packaging has risen in popularity as a viable alternative to synthetic packaging and non-biodegradable petroleum-based packaging in order to reduce packaging’s environmental effects [130]. PBS has outstanding biodegradability and biocompatibility. Its highly transparent surface and rigid construction allow it to be used in a wide variety of applications, including mulching films, compostable bags, nonwoven sheets and textiles, catering goods, and foams [33,131]. Other researchers have stated that PBS is widely found in industries such as agriculture, fishery, forestry and civil engineering. PBS is also used for vegetation nets in the agricultural industry [29]. PBS also can be applied to monofilament, injection molded products, tape, split yarn and textiles industries [132]. Figure 15 shows a variety of PBS applications.

### 6.1. Biomedical

In the biomedical area, PBS is used in various ways. According to Gigli et al. [34], PBS has high potential in making bone marrows stem cell. The results of PBS have showed a higher tendency compared to the PLA and PolyVinyl Chloride (PVC). PBS can be applied in bone tissue engineering which can generate new tissue growth, however, it has limitations due to insufficient osteoblast compatibility and bioactivity [61]. Li et al. [132] studied the degradation behavior of PBS; physiological environments played an important role in the engineering process of a new tissue which affects cell growth, tissue regeneration and host response. The degradation rate showed its potential application as a biomaterial for tissue repair and tissue engineering. Tissue engineering needs good properties in terms of mechanical and molecular weight for tissue replacement. In addition, the degradation rate is also important to ensure the culture can absorb in tissue structures [133]. Other researchers have studied blending PBS with inorganic materials, which can enhance mechanical, thermal, spherulite size and gas permeability which concern the development of tissue engineering. Other than that, it helps to enhance cellular interactions, such as selective endocytosis, adhesion and orientation to stimulate damage tissue [134]. A study on blending PBS with chitosan showed an increase in antimicrobial, antitumor activity, improved protein absorption and rapid cell growth compared with neat PBS [135,136]. High compressive strength is needed for tissue engineering study, as well as the ability to enable cell attachment, proliferation, and extracellular matrix (ECM) deposition, both of which lead to in vivo bone regeneration with a suitable degradation rate.

However, the development of tissue engineering from PBS has a few challenges because PBS is disposed to bacterial infection and insufficient osteocompatibility after implantation into the body, thus one needs to improve the properties by surface treatment or modification. Extensive research also has been carried out which uses a PBS copolymer for the preparation of scaffolds that enhance the regeneration of bone in the dental socket. This material allows bone cells to move in and supports their attachment to the construct simultaneously [137].

### 6.2. Food Packaging

Polymers are widely use in packaging industries. In Europe, over 60% of waste plastic coming from packaging. As a consequence, PBS is a biodegradable material with uses in food packaging application to protect the environment and food quality [138]. A research based on blend PBS nano fibril with PLA improved strength, flexibility and ductility which are important factors in developing food packaging [139]. Hassan et al. [140] claim that the PBS is suitable for food packaging because it has high flexibility, good resilience, high elongation at break, lower glass transition temperature, and good biodegradability. Due to its low molecular weight, low melting point (114 °C), and low stiffness and power, its possible applications are restricted. As a result of his research, blending PBS with PLA improved the properties of the substance for use in food packaging [141] studied, blending PBS with chitosan powder can contribute to food preservation and shelf life extension. Antimicrobial effects are significant in food packaging because they contribute to the barrier properties of the products. Chitosan is a type of bioactive polysaccharide that has antimicrobial properties. Active packaging properties are a form of packaging that alters the state of the packaging in order to prolong shelf life or increase protection while preserving food quality. For the European FAIR-project CT 98-4170, this concept of active packaging was selected. Active packaging also has unusual properties that are not found in standard packaging [142]. PBS was used as a direct-melt coating for paperboard in two different ready-to-eat convenience food packages which are able to reheat in the microwave. PBS also was coated to prevent moisture and grease in convenience for food packaging applications [22]. From the point of view of Valentina et al. [143], not only are mechanical properties critical, but so is food compatibility, which has been established as a possible cause of food quality degradation. During the manufacture of food packaging, it is important to recognize any modifications in the bio plastics’ characteristics that can arise during their contact with food. Unfortunately, the use of biodegradable packaging is restricted due to natural polymers’ low barrier properties and sluggish mechanical properties. To satisfy the specifications of food processing, it must be combined with other polymers or cross-linked agents. Lastly, much research was carried out to produce food packaging which is biodegradable and low-cost [144].

### 6.3. Mulch Film

Modern agriculture heavily relies on the use of conventional plastic mulch films, because these films can raise crop yields through elevating soil temperatures, conserving soil moisture, controlling weed growth and providing protection against severe weather and pests. Mulch films are used for covering the soil to make more favorable conditions for plant growth, development and efficient crop production. The global agricultural film market is predicted to reach an annual volume of 7.5 million tons by 2021, and China uses the most polyethylene (PE) mulch film, with 1.5 million tons annually. After these PE mulch films have been used up, it is hard to recover them from agricultural fields completely, due to PE film embrittlement and fragmentation caused by weathering, particularly when thin films are used [145]. Low-density polyethylene (PE) is the most commonly used plastic mulch because it is inexpensive, easily processed, highly durable and flexible. However, due to environment concerns, common bio-based polymers have been used in mulch films including polylactic acid (PLA), starch, cellulose, and polyhydroxyalkanoates (PHA). Fossil-sourced polyesters used in mulch films include poly(butylene succinate) (PBS), poly(butylene succinate-*co*-adipate) (PBSA), and poly(butylene-adipate-*co*-terephthalate) (PBAT) [146]. Polymers used in biodegradable plastic mulch contain ester bonds or are polysaccharides, which are amenable to microbial hydrolysis [147]. Biodegradable plastic such as PBS can be degraded by microorganisms in the natural environment. Commercially available PBS mulch film is the main component and are blended with various other biodegradable polymer to control their strength and degradability speed adequate for “self-destructs” after their useful lives have ended. The degradation speed was influenced mainly by the surrounding environment [148]. This has been reported by a study on the properties of mulch film prepared from PBS with natural sorbent and fertilizer. Producing thin mulch films with the addition of superabsorbent polymer (SAP) blends loaded with ammonium sulfate ((NH_4_)_2_SO_4_) as a source of fertilizer improved the moisture sorption of the films. This research was showed to improve the films’ rigidity and accelerate the biodegradation of mulch film [149]. In Japan, cultivated fields soil used poly(butylene succinate-*co*-adipate) (PBSA) in mulch films to reduce the removal labor and environmental impact of used film waste. Microorganisms in the soil act to degrade the mulch films and finally convert them into water and CO_2_ or methane in the environment [150]. Figure 16 shows the mechanism on the biodegradation of plastic by microorganisms.

### 6.4. Tableware

Disposable tableware usually is made from synthetic plastic which are commonly found. In Taiwan, there is about 400 night markets and many shops that use much of this cutlery such as disposable chopsticks, foam bowls, and other disposable tableware which is difficult to degrade and not environmentally friendly [152]. There was a study on the potential biodegradable polymer which can be used to substitute current synthetic plastic for tableware. Recent research on bio plastic from starch-based materials for tableware application was studied. It was showed that the starch-based bio plastic was at the optimum biodegradation rate when expose to temperatures at about 105 °C [142]. Among other bio polymers, PBS holds the most promise as an industrially applicable biodegradable material. Moreover, it is a competitive biodegradable plastic material because of its good process ability, high chemical resistance, availability, low material cost, and excellent mechanical and thermal properties which suit tableware application [153]. However, the PBS application for tableware is still rarely used in worldwide commercial settings due to the high cost of the raw material.

## 7. Techno Economic Challenges of PBS

The production of PBS could have a greater impact on worldwide plastic production. Therefore, the techno economic challenge of PBS production must be given major attention before it successfully replaces current synthetic plastic and is used as a future reference. The primary criteria that should be taken into consideration is cost and availability. PBS initially has to achieve its economic and technical targets in the laboratory and is then scaled up to an industrial scale with designed metrics within the complete system. The bulk price of PBS was estimated as average values from different suppliers on www.alibaba.com (11 February 2021) at around USD 50/kg. Besides the costs, the revenues generated from the process will result in profits or loss of the industry. Intuitively, to be able to gain profits, the unit price of the plastic sold must be greater than the unit price of the production. Hence, it depends on the amount of the plastics (production volume) that can be sold in a year. Other than that, for producing rigid products from PBS, the material needs some modification and additive which could overcome the brittleness properties of PBS [154]. Various grades of PBS also may be taken into consideration during the selection of material for a specific application and manufacturing process. The availability of PBS is limited, especially in Malaysia. Therefore, the cost for import raw materials is accountable when producing a product from PBS.

In order to overcome environmental issues, PBS is one of the biodegradable polymers that is suitable for a wide range of applications to substitute current synthetic polymers. However, due to less awareness on the impact of synthetic polymers, it has not been delivered to the consumer [155]. Therefore, the minimum demand for biodegradable polymers may cause a high price in market. It was suggested to overcome this matter through government involvement and an environmental campaign which can educate people on the importance of using biodegradable polymers. The government also can invest in biodegradable polymer research in local universities to encourage industry to produce biodegradable products. Other than that, researchers also may research on waste treatment technologies for biodegradable plastics because it is mainly driven by the situation under which biodegradable plastic products are difficult or non-recyclable in an economically viable way [156]. Biodegradable polymers will compost naturally in soil for a certain period of time. Due to this, many industrial composting facilities tend to decline plastic bags as organic waste bags due to the high capital and operation cost.

To achieve the benefit of biodegradable plastics, it is necessary to adopt them to a scale big enough for the change. Obstacles imposed by improper or fragmented policies and regulations should first be identified, modified or removed.

## 8. Conclusions

Concerns over plastic dumps and waste have driven attempts to create biodegradable composites. Synthetic fabrics and petroleum-based polymers, which are impossible to recycle and degrade, may be supplemented with PBS and natural fibers. PBS composites filled with natural fibers have a wide variety of uses. Natural fiber-reinforced composites have a wide range of features.

This biocomposite material, which have a range of appealing properties, could soon be competitive enough to replace conventional synthetic materials derived from fossil fuels. However, efforts to produce entirely biodegradable composites made of natural fibers and biopolymers that can totally replace synthetic fibers and polymer-based composites have so far met with modest success.

PBS and natural fiber-based PBS composites have undergone comprehensive studies to further understand their behavior and properties. The majority of these experiments reported similar results, supporting previous observations that natural fibers and biopolymers complement each other in a number of ways, although some laboratory studies showed a decline in properties. In contrast to synthetic polymers and fiber products, the increasing abundance of biopolymers, the special properties of natural fibers, and the environmentally sustainable quality of biodegradable plastic products warrant more research into producing PBS-based natural fiber composites. However, in order to effectively substitute synthetic polymer composites with entirely biodegradable composites, a paradigm shift in the synthesis and processing of PBS will be needed for further improvements of PBS-based natural fiber composites.

PBS-based natural fiber composites are expected to have mechanical, practical, and biodegradability properties comparable to synthetic composites in the future. The following are future growth developments in PBS and its composites: first, low-cost manufacturing that can attract mainstream recognition. The cost would most likely decrease as consumer demands, mass manufacturing of biocomposites, and the availability of cheaper biopolymers increase. Second, current and future research can concentrate on the fabrication and improvement of PBS-based composites for multifunctional applications using various varieties, ratios, and shapes of natural fibers. Finally, due to the complex and varied existence of natural fibers, a proper library on fibers and biocomposites should be planned. In the near future, completely biodegradable composites with outstanding multifunctional properties would be feasible. Fiber alteration techniques, such as alkali, silane, binding agents, and other chemicals, can enhance fiber surface properties and the fiber/matrix interface, resulting in improved biocomposites to satisfy a range of specifications. In addition, future research studies should concentrate on the integration of nanocellulose and/or nanoclay into biocomposites, which can improve the different functional properties, as well as the analysis of the tribological properties of PBS-based natural fiber composites.

The flexibility will limit the applications of pure PBS, however, the disadvantage can be overcome by blending it with natural fiber, through the copolymerization process and modification. Blending with natural fibers will reduce the price of PBS products and copolymerization will produce rigid PBS plastic for various applications. With respect to the final application, the studies demonstrated the potentiality of PBS to be employed in different areas of medical ranging, food packaging, mulch films and table ware. A wide-range application using PBS will attract people to the use of PBS in the future.

## Figures and Tables

**Figure 1 polymers-13-01436-f001:**
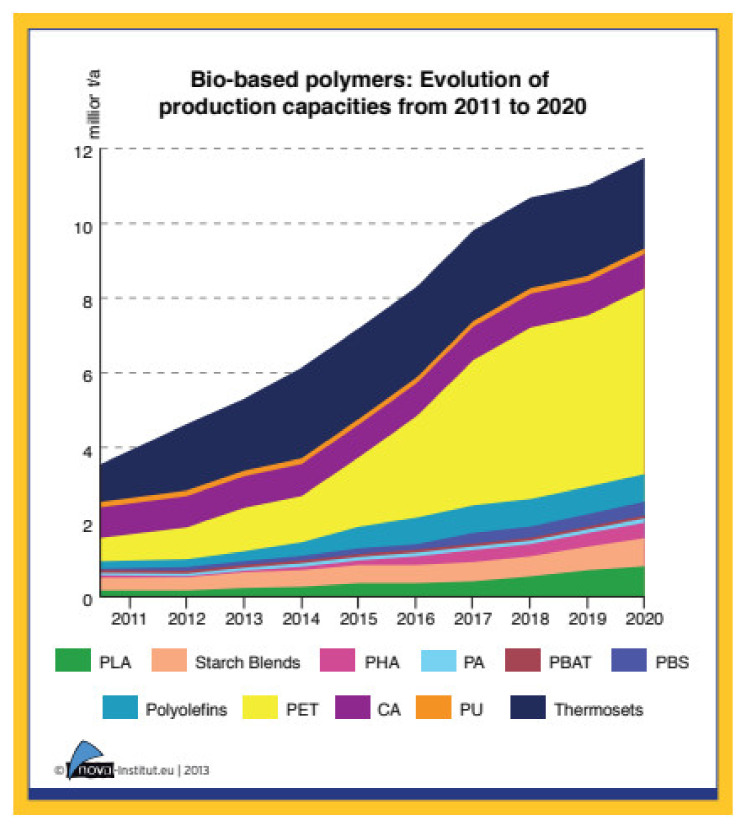
Bio-based polymers production capacities from 2011 to 2020 [14]. PLA: polylactic acid; PHA: polyhydroxyalkanoate; PA: polyamide; PBAT: poly(butylene-adipate-co-terephthalate); PBS: poly(butylene succinate); PET: polyethylene terephthalate; CA: Cellulose acetate; PU: polyurethane. Adapted with permission from Aeschelmann and Carus (2015).

**Figure 2 polymers-13-01436-f002:**
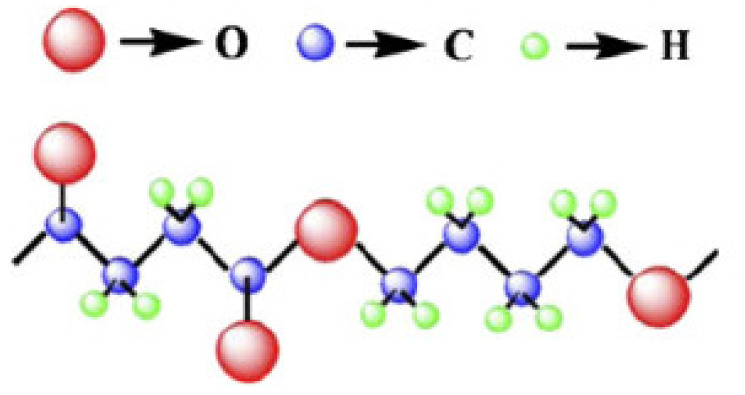
Chemical structure of poly butylene succinate [42]. Adapted with permission from AKanemura et al. (2012).

**Figure 3 polymers-13-01436-f003:**
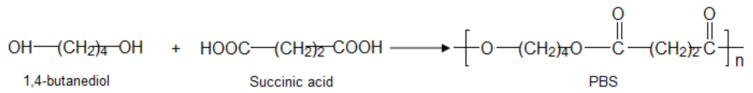
Synthesis of PBS [43]. Adapted with permission from Yu et al. (2011).

**Figure 4 polymers-13-01436-f004:**
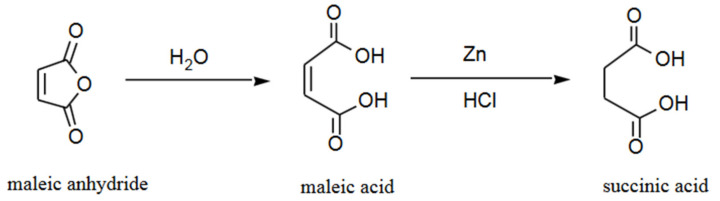
Reaction of maleic anhydride to succinic acid [45]. Adapted with permission from Αδαμοπούλου (2013).

**Figure 5 polymers-13-01436-f005:**
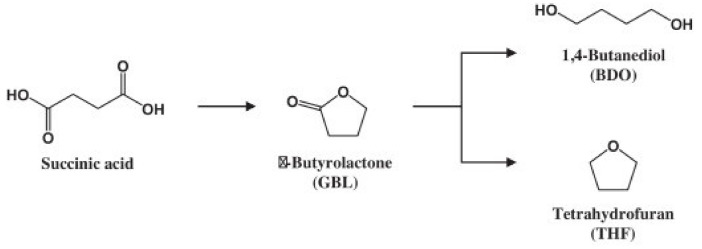
Reaction of succinic acid into 1,4-butanediol to produce PBS [46]. Adapted with permission from Delhomme (2009).

**Figure 6 polymers-13-01436-f006:**
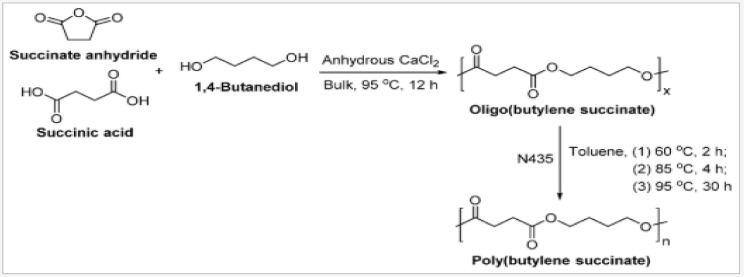
Synthesis of poly(butylene succinate) via the N435-catalyzed co-polymerization of succinic acid and 1,4-butanediol with succinate anhydride [50]. Adapted with permission from Azim et al. (2006).

**Figure 7 polymers-13-01436-f007:**
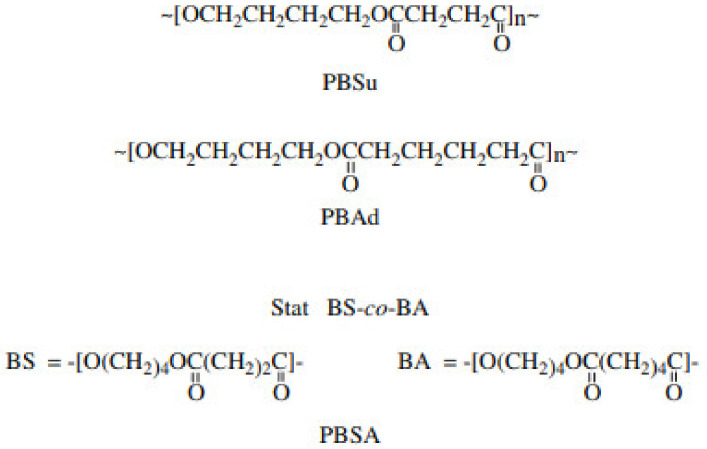
Structure of homopolyesters poly(butylene succinate) (PBSu), poly(butylene adipate)(PBAd) and copolyesters poly(butylene succinate-co-butylene adipate) (PBSA) via polycondensation process [55]. Adapted with permission from Díaz et al. (2014).

**Figure 8 polymers-13-01436-f008:**
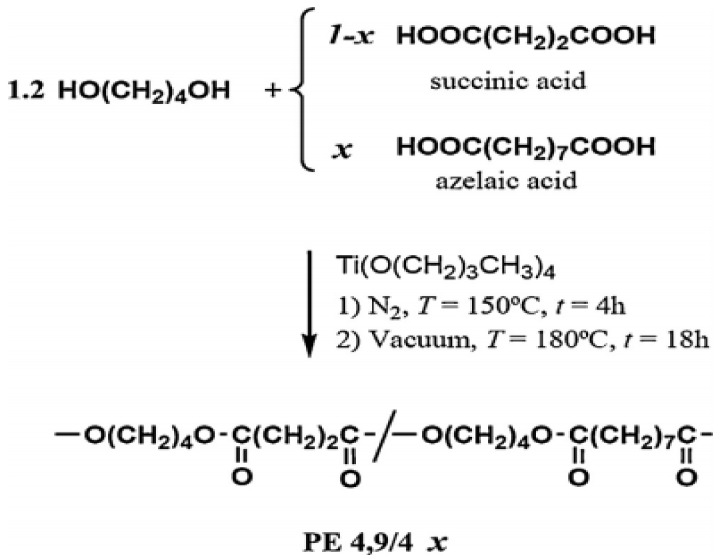
Synthesis of poly(butylene azelate-co-butylene succinate) copolymers [56]. Adapted with permission from Park et al. (1998).

**Figure 9 polymers-13-01436-f009:**
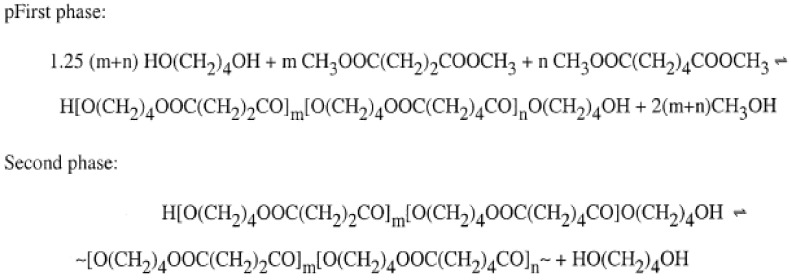
Synthesis of the aliphatic copolyesters [49]. Adapted with permission from Jiang and Loos (2016).

**Figure 10 polymers-13-01436-f010:**
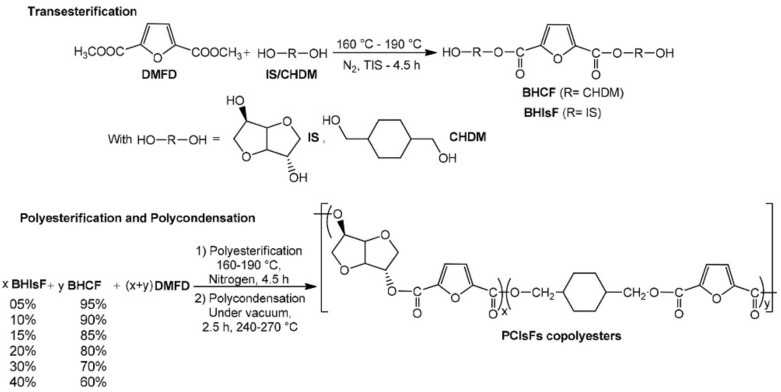
Synthesis of the aliphatic copolyesters of poly(1,4-cyclohexanedimethanol-co-isosorbide 2,5-furandicarboxylate) [47]. Adapted with permission from Kasmi et al. (2018).

**Figure 11 polymers-13-01436-f011:**
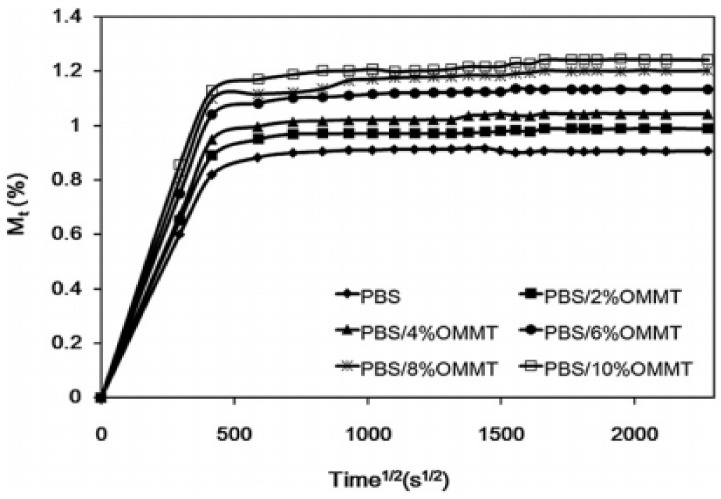
Typical moisture uptake curves at 30 C and 90% RH [72]. Adapted with permission from Nam et al. (2012).

**Figure 12 polymers-13-01436-f012:**
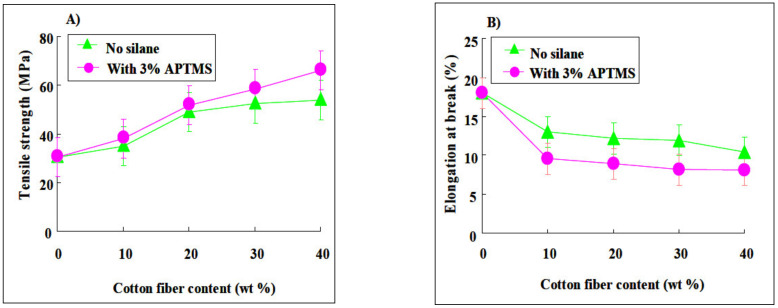
(**A**)Tensile strength and (**B**) elongation at break of PBS/cotton fiber composite [8]. Adapted with permission from Calabia et al. (2013).

**Figure 13 polymers-13-01436-f013:**
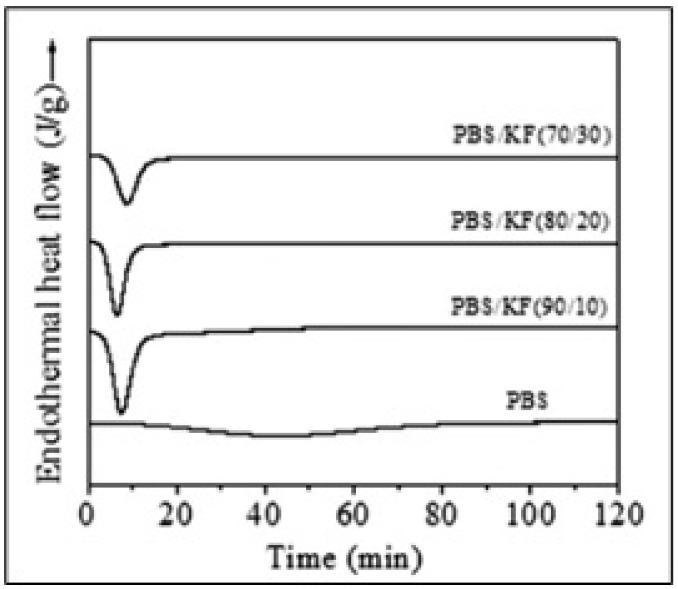
DSC curves of PBS and PBS/kenaf fiber (KF) composites isothermally melt-crystallized at 100 °C [86]. Adapted with permission from Pinho et al. (2009).

**Figure 14 polymers-13-01436-f014:**
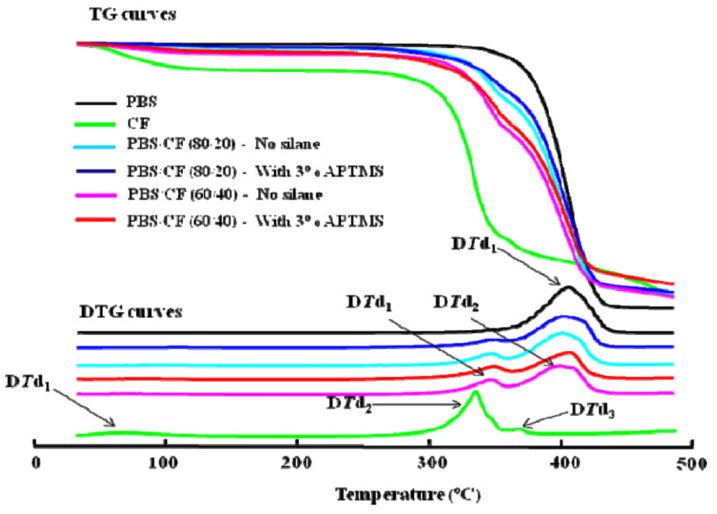
Thermogravimetry (TG) and derivative thermograms (DTG) curves of PBS, chopstick hybrid fiber (CF), PBS/CF (80/20 wt%), and PBS/CF (60/40 wt%) composites [8]. Adapted with permission from Calabia et al. (2013).

**Figure 15 polymers-13-01436-f015:**
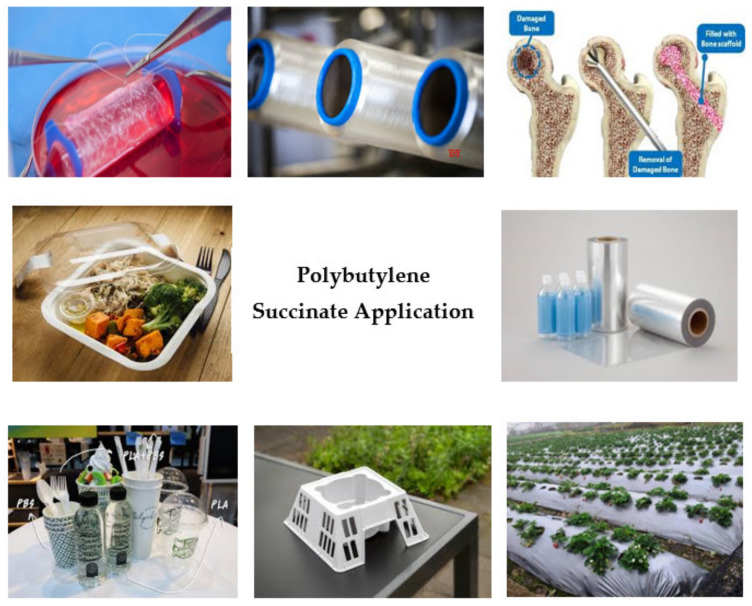
PBS Application.

**Figure 16 polymers-13-01436-f016:**
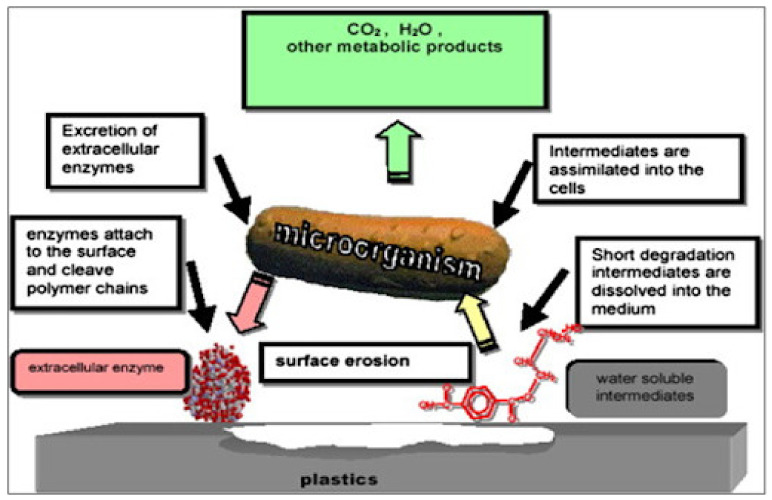
Mechanism plastic degradation by microorganism [151]. Adapted with permission from Sun and Lin (2019).

**Table 1 polymers-13-01436-t001:** Properties of different grades of PBS. T_m_: melting temperature; T_g_: glass transition temperature.

PBS Grade	Company	MFI	Density (g/cm^3^)	Tg (°C)	Tm (°C)	Tensile Strength (MPa)	Tensile Modulus (MPa)	Reference
Molecular Weight 80,000	Anqing Hexing chemical co.ltd	-	1.24	−44.3	109	27	-	[35]
Bionelle 1020 MD	Showa Denko (Tokyo, Japan)	20–34 g/10 min (at 140 °C and 2.16 kg)	1.23	59.7	114.1	-	643	[36]
Bionelle 1020 MD	Showa Highpolymer (Tokyo, Japan)	25 g/10 min at (190 °C, 2.16 kg)	1.26	−32	115	33.7	707	[37]
FZ91PM	PTT Public Company Limited in Thailand.	5 g/10 min at (190 °C, 2.16 kg)	1.26	78	115	20	450	[17]
Bionelle	HKH National Engineering Research Centre of Plastic	5 g/10 min at 150 °C, 2 kg	1.26	−32	114	32	30	[38]

**Table 2 polymers-13-01436-t002:** Properties of blended PBS with different copolymers.

Copolymer + PBS	Properties	References
Hexamethylene	- Increase crystallinity - Increase degradation rate	[30]
Ethylene	- Improve surface tension - Reduce crystallinity - Increase degradation rate	[58]
1,6-hexanediamine	- Increase wettability	[59]
Thiodiglycolate	- Improve thermal stability - Decrease molecular weight - Enhance surface hydrophilicity	[59]
Thiodiethylene succinate	- Increase crystallinity - Increase degradation rate	[7]
Terephthalate	- Improve strength - Improve thermal stability	[60]

## Data Availability

No Data available.

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
