# Peer review of "A Review on Properties and Application of Bio-Based Poly(Butylene Succinate)"

_polymers, 2021, doi:10.3390/polym13091436_

Round 1

Reviewer 1 Report

I read carefully the review article entitled ‘Poly (butylene succinate): An approach towards biodegradability’. The concept of the review article is interesting but there are many lacking points. This manuscript needs substantial revision for improvement and to be published in the reputed Polymers Journal.

The major comments as follows:

  • Title should be modified in the precise way. Only biodegradability is important or other properties are also influenced in PBS commercial or  practical application.
  • Abstract section should be rewritten and look very general and not informative. In abstract authors should mention the importance of the selection of PBS briefly.
  • Provide a nice graphical abstract representing the overview of the MS with key highlights. For review articles it should be mandatory and don’t use the figures used in the manuscript.
  • Introduction looks very general. In the introduction section, write the novelty of the work and the problem statement clearly.
  • Detailed discussion about quantitative data of commercial plastics and their toxicity to the various ecosystems need to be addressed. For this refer and cite some recent review artciel Bioresource Technology Volume 325, April 2021, 124685. For line no 62 refer and cite PHB production using  lignocellulosic biomass Polymers 12 (8), 1704, 2020. Author should discuss more information about the commercial production of PBS worldwide and their data in the introduction section.
  • Regarding synthesis and physical and chemical properties this review looks similar with previous reported review article author should consider and give proper justification to this comment during revision.
  • Authors used a cluster of references many times. It is highly recommended to avoid this and give all details of each reference during revision. 
  • Give more details on the application of PBS add one or two figure for the same.
  • Techno Economic challenges of PBS in practical applications and future research perspectives and challenges need to be addressed by adding a new section before conclusions.
  • What are the limitations to use PBS  for commercial application?
  • The conclusion of the study is not discussed with the specific output obtained from the study, it could be modified with precise outcomes with a take home message.
  • English and grammar mistakes are present. The author should check the manuscript by native English Speaker to improve the quality of the manuscript.

Author Response

The major comments as follows:

  • Title should be modified in the precise way. Only biodegradability is important or other properties are also influenced in PBS commercial or practical application.

After our discussions, we decided to modify our title into: A review on Properties and Application of Bio-based Poly (butylene succinate)

  • Abstract section should be rewritten and look very general and not informative. In abstract authors should mention the importance of the selection of PBS briefly.

The abstract has been revised and informative.

  •  Provide a nice graphical abstract representing the overview of the MS with key highlights. For review articles it should be mandatory and don’t use the figures used in the manuscript.

The graphical abstract was added below abstract.

  • Introduction looks very general. In the introduction section, write the novelty of the work and the problem statement clearly.

The introduction section has been revised into more precise and novelty.

  •  Detailed discussion about quantitative data of commercial plastics and their toxicity to the various ecosystems need to be addressed. For this refer and cite some recent review artciel Bioresource Technology Volume 325, April 2021, 124685. For line no 62 refer and cite PHB production using  lignocellulosic biomass Polymers 12 (8), 1704, 2020. Author should discuss more information about the commercial production of PBS worldwide and their data in the introduction section.

The introduction section has been revised into more informative.

  •  Regarding synthesis and physical and chemical properties this review looks similar with previous reported review article author should consider and give proper justification to this comment during revision.

After our discussions, we found that the synthesis part is hardly change because the synthesis has fix and we are not focus on innovative synthesis. But the physical and chemical properties were revised.  

  •  Authors used a cluster of references many times. It is highly recommended to avoid this and give all details of each reference during revision. 

The references were revised.

  •  Give more details on the application of PBS add one or two figures for the same.

The PBS applications figures have been added into manuscripts.

  •  Techno Economic challenges of PBS in practical applications and future research perspectives and challenges need to be addressed by adding a new section before conclusions.

The challenges section has been added into manuscript

  • What are the limitations to use PBS for commercial application?

The limitations discussion has been added into introduction section.

  • The conclusion of the study is not discussed with the specific output obtained from the study, it could be modified with precise outcomes with a take home message.

The conclusion has been revised.

  • English and grammar mistakes are present. The author should check the manuscript by native English Speaker to improve the quality of the manuscript.

English proofreading was done on this manuscript.

Reviewer 2 Report

  1. Line 35, page 1, it is mentioned that "Plastic expanded......... over the span 1950 to 2012......." why authors provide data till 2012 in 2021.
  2. Line 42, rewrite km2 to km2, line 231, rewrite H2O or NH3, authors need to proofread the paper with great care. 
  3. Line 396 and caption of figure 14 author need to check degree C; they put different at different places (for reference, check degree C at line 399)
  4. Line 55 provides the full name of BCC followed by an abbreviation.
  5. The current form of the article looks like a report. The authors just presented the other's work. They need to analyze the literature critically. 
  6. They should discuss the prospect of the topic. 
  7. The authors should put more applications why they only focus on two applications as the title is general, then no point in limiting the applications. 
  8. The literature presented is quite old, i.e., before 2015. No recent trends have been discussed, and hence, the reference statistics appear to be quite strange. The authors have presented the work in 2021 and have not taken a single reference from 2020. Only 16% of the review material is from the last five years. Therefore, the article is not worthy enough to highlight the recent developments of the current topic.
  9. The authors should check this paper https://doi.org/10.3390/polym12071571 in the same journal in 2020. There are many recent papers published on this topic.
  10. The authors need to modify the figures with better resolution.

Author Response

  • Line 35, page 1, it is mentioned that "Plastic expanded......... over the span 1950 to 2012......." why authors provide data till 2012 in 2021.

The data has been updated up to forecasting 2025.

  • Line 42, rewrite km2 to km2, line 231, rewrite H2O or NH3, authors need to proofread the paper with great care. 

The manuscript has been rechecked and revised.

  • Line 396 and caption of figure 14 author need to check degree C; they put different at different places (for reference, check degree C at line 399)

The manuscript has been rechecked and revised.

  • Line 55 provides the full name of BCC followed by an abbreviation.

The full name of BCC, Business Communication Company has been added into manuscript.

  • The current form of the article looks like a report. The authors just presented the other's work. They need to analyze the literature critically. 

The manuscript has been revised.

  • They should discuss the prospect of the topic. 

After our discussions, we decided to modify our title into: A review on Properties and Application of Bio-based Poly (butylene succinate)

  •  The authors should put more applications why they only focus on two applications as the title is general, then no point in limiting the applications. 

Multiple application sectors have been added and discussed in this manuscript.

  • The literature presented is quite old, i.e., before 2015. No recent trends have been discussed, and hence, the reference statistics appear to be quite strange. The authors have presented the work in 2021 and have not taken a single reference from 2020. Only 16% of the review material is from the last five years. Therefore, the article is not worthy enough to highlight the recent developments of the current topic.

The citation references have been revised

  • The authors should check this paper https://doi.org/10.3390/polym12071571 in the same journal in 2020. There are many recent papers published on this topic.

The citation references have been revised

Round 2

Reviewer 1 Report

The authors have substantially revised the manuscript according to the comments.

The present form of the manuscript can be accepted for publication.

Author Response

Thank you for your valuable time to review our manuscript. 

Reviewer 2 Report

  1. The authors addressed all raised comments satisfactorily.  But need the following minor corrections:
    1. In section 7 authors need more references about the Techno-Economic Challenges. Only alibaba.com and one more reference are not enough to explain the important point.

Author Response

In section 7 authors need more references about the Techno-Economic Challenges. Only alibaba.com and one more reference are not enough to explain the important point

  • Further elaboration and discussion were done in this section.